# The Leaf Microbiome of Tobacco Plants across Eight Chinese Provinces

**DOI:** 10.3390/microorganisms10020450

**Published:** 2022-02-16

**Authors:** Haiyang Hu, Yunli Liu, Yiqun Huang, Zhan Zhang, Hongzhi Tang

**Affiliations:** 1State Key Laboratory of Microbial Metabolism, School of Life Sciences and Biotechnology, Shanghai Jiao Tong University, Shanghai 200240, China; liuyunli@sjtu.edu.cn (Y.L.); natsumoon@outlook.com (Y.H.); tanghongzhi@sjtu.edu.cn (H.T.); 2China Tobacco Henan Industrial Co., Ltd., Zhengzhou 450000, China; zhangzhan2059729@126.com

**Keywords:** crop microbiota, leaves, core bacterial community, industrial factors, environmental detoxification

## Abstract

Leaf microorganism communities play significant roles in the process of plant growth, but the microbiome profiling of crop leaves is still a relatively new research area. Here, we used 16S rDNA sequencing to profile the microbiomes of 78 primary dried tobacco leaf samples from 26 locations in eight Chinese provinces. Our analyses revealed that the national leaf microbial communities contain 4473 operational taxonomic units (OTU) representing 1234 species, but there is a small, national core microbiome with only 14 OTU representing nine species. The function of this core microbiome is related to processes including nitrogen fixation, detoxification of diverse pollutants, and heavy-metal reduction. The leaf microorganism communities are obviously affected by local environments but did not exhibit obvious relationships to single ecological factors (e.g., temperature, precipitation). Our findings enhance the understanding of microbial diversity of tobacco leaves, which could be utilized for a variety of bioprocess, agricultural, and environmental detoxification applications.

## 1. Introduction

With the help of recently developed techniques for culture-independent microbial identification, the microbial diversity and distribution of various environments can now be readily studied, including the human gut, wastewater treatment plants, and soils [1,2,3]. Such studies are helpful for improving our understanding of bioprocesses occurring in these environments and our understanding of risks to human health, and these studies also facilitate the harnessing of microorganisms for human use [4]. Of particular note, both Jizhong Zhou and Noah Fierer have recently organized global-sampling projects, respectively collecting 1200 activated sludge samples from 23 countries on six continents and 237 soil samples from six continents [5,6]. These studies yielded “most wanted” lists for future experimental efforts to understand the microbiome of activated sludge and soil, and it is clear that such efforts can greatly promote microbial resource development in related fields.

Studies of multiple crop species have shown that microorganisms resident on the surface of plant leaves (e.g., *Populus trichocarpa*, *Arabidopsis*, *Nicotiana tabacum* L.) alter the chemical composition of leaves, promote crop growth, and confer antibacterial effects [7,8,9,10]. Nevertheless, leaf profiling of crop species is a relatively new research area, so there are many gaps in our knowledge, including the variations in the basic microorganism compositions and the factors that can influence the function of particular taxa on leaves. *Bacillus subtilis* and *Bacillus cricosporum* were isolated from fermented leaves in 1967, both of which release pleasant fragrance compounds [11], and *Enterobacter cloacae* was isolated from the surface of tobacco leaves in 1983 and showed that it degrades nicotine [12].

More recently, Mingqin Zhao successfully utilized a mixture of four dominant bacteria alongside *α*-amylase and protease treatments during leaf fermentation to reduce irritant gas production [13,14]. Advancing this research area beyond screening microorganisms using conventional media culture, Jingwen Huang and Can Su applied 16S rDNA sequencing to analyze the bacterial communities of unaged and aging flue-cured tobacco of the ‘K326’ and ‘Zimbabwe’ cultivars [15,16]. There have also been studies of the microbial diversity of tobacco in China, including analysis of samples from Fujian Province [17,18]. These studies have discovered some new microorganisms, including *Sphingomonas*, *Enterobacter*, and *Pantoea*, among others, which could not have been detected using traditional culture-based methods.

In the present study, we profiled the microbiomes of 78 primary dried tobacco leaf samples from plants grown at 26 locations in eight Chinese provinces. We systematically analyzed the leaf microbial community across the country and obtained a “core microbiome” for the whole country and core microbiomes for each of the eight provinces. We also analyzed the predicted functions of the operational taxonomy units (OTUs) in the core microbiomes. Our findings emphasize that many potentially useful functional bacterial resources are present on tobacco leaves, suggesting how they can be harnessed for a variety of bioprocess, agricultural, and environmental detoxification applications.

## 2. Materials and Methods

### 2.1. Sample Collection

A total of 78 primary dried tobacco leaf samples were collected from 26 locations in 8 Chinese provinces: YN, Yunnan province; HeN, Henan province; CQ, Chongqing city; HuN, Hunan; GZ, Guizhou province; SC, Sichuan province; HLJ, Heilongjiang province; FJ, Fujian province. The primary dried tobacco leaves were taken from the storage of sampling locations at the same time and transferred to the lab. We stored them at 4 °C and handled them within a month. The detailed information of the sampling locations, longitude, latitude, annual average temperature, annual precipitation, and altitude were listed in Appendix A.

### 2.2. Sample Treatment

A whole crop leaf from the collected sample was settled in a sterilized flask with 300 mL sterilized 1 M phosphate buffer (pH 7.0). The flask was further shaken at 16 °C for 30 min. The supernatant was filtered with four layers of sterilized gauzes to remove residue and then re-filtered with a 0.22 μm filter membrane to collect microorganisms. The membranes carrying gathered microorganisms were transferred to a sterile centrifugal tube, frozen in liquid nitrogen for 15 min, and finally stored at −80 °C [19].

### 2.3. 16S rDNA Gene Sequencing and Sequence Processing

The samples were sent to Shanghai Personal Biotechnology Co., Ltd. for DNA extraction and further sequencing. The primers 27F (5′-AGAGTTTGATCMTGGCTCAG-3′) and 1492R (5′-ACCTTGTTACGACTT-3′) were applied to amplify 16S rDNA [20].

Pacino Sequel platform (the third-generation sequencing platform) was applied to sequence the full-length 16S rDNA of the community (https://github.com/PacificBiosciences/smrtflow, accessed on 1 February 2022). To ensure the reliability and accuracy of the analysis results, QIIME software (Quantitative Insights Into Microbial Ecology, v1.8.0, http://qiime.org/, accessed on 1 February 2022) was used to exclude uncertain sequences [21]. Sequences with 5′ mismatched base number >5 and with continuous same base number >8 were removed. Then, we used USEARCH (v5.2.236, http://www.drive5.com/usearch/, accessed on 1 February 2022) to check and remove the chimera sequence and obtain 945,434 processed sequences.

### 2.4. Sequence Comparison against Reference Databases

UCLUST, a sequence alignment tool, was used to merge the 945,434 processed sequences and demarcate OTUs according to 97% sequence similarity [22]. The sequences with the highest abundance in each OTU were selected as the representative sequences of this OTU. The representative sequences of OTU were compared with the NCBI database to obtain the taxonomic information.

### 2.5. Species Abundance Distribution (SAD) Fitting

Species abundance distribution is a description of the abundance (number of individuals observed) for each different species encountered within a community; this can also be described as a community-level metric denoted by Φ(n) that tells us the probability of a given species having “n” abundance [23]. Thus, SAD is one of the major tools in ecology [24].

### 2.6. Core Bacterial Community Determination

OTUs meeting all three criteria were defined as core bacterial community. First, the mean relative abundance of core OTUs must be more than 1% in every sample. Second, core OTUs should be identified in more than 80% of all samples. Third, the OTUs were identified as abundant when they made up the top 80% of the reads in the sample. The core OTUs should be identified as abundant in at least half of all the samples [25,26].

### 2.7. α-Diversity

*α*-diversity indices, including Shannon [27], Simpson [28], Chao1 [29], and ACE [30], were calculated using QIIME software. Chao1 and ACE indices were employed to determine community richness, and Shannon and Simpson indices were used to determine community diversity. Then, we used QIIME and R software (R-3.2.0) to draw the rarefaction curve and species accumulation curve, respectively [5]. By randomly extracting a certain number of sequences from each sample, a rarefaction curve is able to predict the total number of species and relative abundance of each species in the sample at a given sequencing depth. A species accumulation curve is able to judge whether the sample numbers were sufficient or not and estimate the community richness.

### 2.8. β-Diversity

The weighted and unweighted UniFrac distances were calculated to represent the phylogenetic *β*-diversity using the R software. With QIIME software, the unweighted pair-group method with arithmetic means (UPGMA) clustering analysis was performed [31]. PCoA (Principal Coordinates Analysis) is a visualization method to study the similarity or difference of data. After sorting through a series of eigenvalues and eigenvectors, the major eigenvalues were selected. R software was used to draw PCoA diagrams, which could show the differences between different samples [32].

### 2.9. Redundancy Analysis (RDA)

RDA analysis was a typical constrained sequencing method. Multiple linear regression was used to fit the data with several given influencing factors including longitude, latitude, altitude, temperature, and precipitation (Appendix A), and a test was used to determine whether the factor had a significant influence on the bacterial composition or not [33]. We identify “significant difference” if the value of −log10 (*p*) is over 1.3.

### 2.10. Construction of Phylogenetic Tree and Taxonomic Tree

A phylogenetic tree with OTU representative sequences was constructed by the Fast Tree tool [34]. MEGAN (http://www.megasoftware.net/, accessed on 1 February 2022) was used to map the abundance information and taxonomy composition data of OTU contained in each sample to the taxonomic tree provided by NCBI Taxonomy (https://www.ncbi.nlm.nih.gov/taxonomy, accessed on 1 February 2022) so that in a standard classification system, the specific composition of all samples at each classification level was presented uniformly.

### 2.11. LEfSe Analysis

LEfSe is an analysis method based on linear discriminant analysis (LDA). With the Galaxy online platform (http://huttenhower.sph.harvard.edu/galaxy/, accessed on 1 February 2022), the relative abundance of the species-level matrix was analyzed. LEfSe gives the logarithmic score value of LDA > 2 and *p* value < 0.05 [35].

### 2.12. Functional Prediction with PICRUSt

Based on the measured 16S rDNA full-length sequences of microorganisms, the gene functional spectrum of common ancestor genes was deduced. By inferring unmeasured 16S rDNA full-length sequences of the Greengenes database, the gene functional prediction spectrum of the complete spectrum of archaea and bacteria was constructed. These two kinds of 16S rDNA gene sequences were compared, and the “nearest neighbor of the reference sequence” of each sequence was found. According to the rRNA gene copy number of the “nearest neighbor of reference sequence”, the obtained OTU abundance matrix was corrected. The data of bacterial composition were “mapped” to the gene functional spectrum database of KEGG, and the metabolic function of the bacterial community was predicted [36].

### 2.13. Determination of Metals-Inductively Coupled Plasma Mass Spectrometry (ICP-MS)

Preparation for tobacco digestion: We crushed the tobacco leaves and dried them at 60 °C to constant weight. Then, we weighed 0.2 g of tobacco powder into a shake flask and added 10 mL of mixed acid (HNO_3_:HClO_4_ = 4:1) overnight (12 h). The solution was digested by electrothermal digestion until the solution was transparent. Then, 1 mL 1% HNO_3_ (*v*:*v*) was added to dilute to 25 mL. The treated samples were detected by ICP-MS [37].

### 2.14. Methods and Tools Used for Visualization

The data in this study were visualized by the Fast Tree tool [34], R software [5], and Office 2021.

## 3. Results and Discussion

We obtained a total of 945,434 sequences from 78 crop samples at 26 sampling locations distributed across eight Chinese provinces (Figure 1A). Operational taxonomy units (OTUs) were demarcated according to 97% sequence similarity. Finally, a total of 4473 OTUs were identified, and these were classified into 11 phyla, 26 classes, 69 orders, 151 families, 440 genera, and 1234 species. We detected 1399 OTUs per province on average, with a minimum of 620 OTUs for Chongqing (CQ) and a maximum of 2473 OTUs in Yunnan (YN) (Figure 1B).

The classification tree for all samples visualized by GraPhlAn is shown in Appendix A, as well as the relative abundances of the microorganisms at the genus level is shown in Figure 1C (Appendix A shows the community compositions at different taxonomic levels). A total of 440 genera were identified from the 78 samples, among which the top 13 genera (in terms of relative abundance) accounted for 82.09% of all OTUs for the entire dataset (details in Appendix A). Notable genera from among these top 13 included *Salmonella* (4.3%), which is harmful for human health.

### 3.1. Species Abundance Distribution (SAD)

In the present study, we applied common SAD models, including Poisson lognormal, log-series, Broken-stick, and Zipf to predict species abundance distributions of crop leaf bacterial communities. The Poisson lognormal model had the best fit for the SAD among all samples and was able to explain 95.34% of the variation of the crop leaf bacterial SAD, thus outperforming the 84.77% for lognormal, 78.88% for log-series, 84.11% for Zipf, and 10.23% for the Broken-stick model. The fitting result is consistent with the previous reports [5,24].

### 3.2. Functional Prediction with PICRUSt

The abundance distributions showed that between 43.10 and 50.48% of the predicted pathways from the OTU data were of the metabolism type (Appendix A). In more detail, the values of the relative abundance for pathways including “carbohydrate metabolism”, “amino acid metabolism”, “energy metabolism”, “metabolism of cofactors and vitamins”, “xenobiotics biodegradation”, and “metabolism, nucleotide metabolism, and lipid metabolism” were 8.91–10.631%, 8.33–10.27%, 4.60–9.11%, 3.69–5.48%, 2.13–3.90%, 2.75–3.19%, and 2.71–3.62%, respectively (Appendix A).

### 3.3. α-Diversity Analysis of the Crop Samples

We next examined the microbial diversity and community composition by characterizing *α*-diversity. First, we confirmed that both the Shannon rarefaction curve and the species accumulation curve indicated sufficient sequencing depth to informatively reveal the community diversity of the samples (Figure 2A,B). The *α*-diversity indices, including Shannon, Simpson, Chao1, and ACE, were applied to assess the microbial diversity of the samples. In the national scale, the Chao1 and ACE index values were quite high (Appendix A), underscoring the high bacterial richness among the samples. Nevertheless, the values of the Shannon and Simpson index were quite low (Appendix A), suggesting that the relative abundances of the taxa in a given sample differed greatly with other samples. The box plot of Shannon and Simpson index indicated that microbial diversity was highest in Fujian province and lowest in Henan province (Figure 2C,D).

### 3.4. A National Core Bacterial Community

Through the analysis of abundance and frequency data, we can identify the dominant species in each of the in situ environments represented by the 26 sampling sites. To determine the core community, OTUs were defined as abundant when they made up the top 80% of the reads in a sample (when ranked by decreasing OTU abundance) (Figure 3A). When the cumulative read abundance reached 80%, 245 OTUs were observed. The frequency of each OTU was also considered in the core community selection. There are 24 OTUs present in more than 80% of the samples, and these together comprise 85.4% of the relative abundance for the whole dataset (Figure 3B). Following principles for core community selection, we cataloged a national core bacterial community, which included 0.31% of the OTUs (14 OTUs) yet occupied 31.01% (±3.94%) of the total relative abundance of the whole dataset, comprising all 78 samples (Figure 3C,D, Table 1). The national core community consisted of the following genera: *Mastigocoleus* (9.35% ± 6.89%), *Pseudomonas* (4.51% ± 2.43%), *Salmonella* (3.27% ± 6.24%), *Leclercia* (3.31% ± 2.81%), *Methylobacterium* (2.46% ± 1.69%), *Enterobacter* (2.17% ± 2.48%), *Atlantibacter* (2.14% ± 3.77%), *Pantoea* (1.93% ± 2.66%), and *Sphingomonas* (1.88% ± 1.57%).

Among these of the national core bacterial community, *Salmonella enterica* is a well-studied pathogenic bacterium, and it is present in almost all of the samples [38]. The functions of *Methylobacterium goesingense*, *Sphingomonas roseiflava*, and *Sphingomonas aurantiaca* are unclear. *Mastigocoleus testarum* is a kind of cyanobacteria previously implicated in biological erosion [39], and *Atlantibacter hermannii* was reclassified from *Escherichia hermannii* and *Salmonella subterranean* [40], but there is little other information available. The remaining four genera—*Pseudomonas* (*Pseudomonas oryzihabitans, Pseudomonas straminea*), *Leclercia* (*Leclercia adecarboxylata*), *Enterobacter* (*Enterobacter soli*), and *Pantoea* (*Pantoea agglomerans*)—have been implicated to function in processes including nitrogen fixation [41], heavy metal reduction [42], insecticidal activity [43,44], and degradation of PAHs [45], lignin [46], and indole-3-acetic acid [47] (Table 1).

**Table 1 microorganisms-10-00450-t001:** The list of the national core microbial community.

Genus	Species	Function
*Mastigocoleus*	*Mastigocoleus testarum*	Function unknown
*Atlantibacter*	*Atlantibacter hermannii*	Function unknown
*Salmonella*	*Salmonella enterica*	Harmful for human body [38]
*Leclercia*	*Leclercia adecarboxylata*	Degradation of PAHs (pyrene) [45]
*Enterobacter*	*Enterobacter soli*	Degradation of indole-3-acetic acid and lignin [46,47]
*Pantoea*	*Pantoea agglomerans*	heavy metals reduction, nitrogen fixation, insecticidal activity [41,42,43]
*Pseudomonas*	*Pseudomonas oryzihabitans* *Pseudomonas straminea*	Insecticidal activity (root-knot nematode) [44]
*Methylobacterium*	*Methylobacterium goesingense*	Function unknown
*Sphingomonas*	*Sphingomonas roseiflava* *Sphingomonas aurantiaca*	Function unknown

### 3.5. Provincial Core Bacterial Communities

Then, the core bacterial communities for each province were determined using the same method as for the national core bacterial community (Appendix A), and we prepared lists for beneficial bacteria that are specific to each province (Appendix A). For example, in Yunnan province, OTU_4397 was close to *Ochrobactrum anthropi*, which is related to aniline degradation [48], and heavy metal detoxification [49]. *Ochrobactrum lupine* (OTU_8084) was the core community member in Sichuan and Chongqing, and it has been shown to degrade pesticides including chlorothalonil, beta-cypermethrin, and 3-phenoxybenzoic acid [50,51]. *Ochrobactrum pseudintermedium* (OTU_8922), specific to Fujian, is known to degrade PAHs [52]. Moreover, *Acinetobacter* sp. (OTU_12502, OTU_9277, OTU_11315) was among the core bacterial communities of Guizhou, Fujian, Hunan, Chongqing, and Sichuan, and it has been shown to degrade PAHs and nicotine and to detoxify heavy metals [53]. A recently reported cellulose-degrading specy, *Beijerinckia fluminensis* (OTU_2253, OTU_13065), was found in the core bacterial communities of both Fujian and Heilongjiang province [54].

We also identified pathogenic bacteria among the province core communities. For instance, the black rot causal pathogen *Xanthomonas campestris* was in the Yunnan core [55], indicating a specific preventing strategy for better growth in Yunnan province. We also found bacteria with reported links to human diseases, including *Kluyvera intermedia* [56] in the Chongqing core, *Stenotrophomonas maltophilia* [57] in the Heilongjiang core, and *Bordetella petrii* and *Bordetella hinzii* [58,59] in the Fujian core. In terms of both the diversity and abundance of pathogenic bacteria, the Chongqing core community had the largest number (three species, 8.85%).

According to the Shannon and Simpson index, the microbial diversity of southeasterly Fujian province—which has a subtropical monsoon climate characterized by high humidity, high temperature, and abundant precipitation—is the highest. It was notable that Fujian province also has the most diverse resources in terms of beneficial bacteria among the examined provinces. Specifically, the Fujian core bacterial core community contained 12 species of beneficial bacteria, which was a number higher than the other seven provinces (Appendix A). Moreover, we noted that Heilongjiang province was apparently the most suitable environment for the beneficial bacteria; this province is in the extreme far north of China and has a cold temperate climate in some regions and a temperate continental monsoon climate in others. The abundance of beneficial bacteria in its core community is 39.8%, which is the highest among all of the examined provinces.

Recalling that *Bacillus* sp. are known to significantly impact the chemical components of tobacco leaves [11], it was notable that these taxa occupied more than 1% of the relative abundance across all samples of the dataset. However, no *Bacillus* sp. were among the national or provincial core bacterial communities. Of the 78 samples, only 16 had more than 1% relative abundance of *Bacillus.* These 16 samples were distributed in seven provinces, Yunnan, Fujian, Henan, Heilongjiang, Sichuan, Hunan, and Guizhou. So, although these taxa are distributed all over China, these are clearly not evenly distributed in every region (Appendix A). It also bears mention that *Arthrobacter* sp., which are known to employ the pyridine pathway to degrade nicotine, were among the detected taxa for some samples but were not included among the national or provincial core bacterial communities. However, *Pseudomonas* sp. strains, which are usually reported to employ the pyrrolidine pathway [60], is commonly present in the samples.

### 3.6. β-Diversity Analysis

We also conducted *β*-diversity analysis to characterize relationships among the samples. Two types of clustering analysis—PcoA analysis of unweighted Unifrac distance and UPGMA (unweighted pair-group method with arithmetic means)—both showed that the samples tend to cluster together by province, suggesting a significant effect from sampling location on microbial community composition (Appendix A). A multi-response permutation procedure was applied to determine the composition difference between each province. More than half of the groups showed significant differences, again indicating that the microbial community composition varies greatly from province to province (Figure 4A).

### 3.7. Redundancy Analysis (RDA) and Correlation with Environmental Factors

We employed RDA to explain the composition difference and to determine the influences of local environmental factors on microbial community composition, including temperature, precipitation, altitude, longitude, and latitude (Appendix A). The microbial community composition of the samples in Yunnan province was most strongly affected by altitude, whereas that of Henan province was most strongly affected by latitude. However, none of the single environmental factors showed a significant effect on microbial communities at the national scale (Figure 4B). The latitudinal-diversity gradients for all samples were also analyzed, but no obvious influence was detected for latitude or longitude, which is consistent with previous reports of microbial latitudinal diversity gradients [61] (Appendix A).

### 3.8. Locally Specific Factors Drive Community Compositions

Our data support that the microbial communities are obviously affected by local environments but did not exhibit obvious relationships to single ecological factors. Thus, we screened the significantly different species between different sampling locations with LEfSe (linear discriminant analysis coupled with effect size) analysis attempting to unravel key factors influencing microbial community structure. This identified eight species in Mangdui, Menglijiaojidi, Qujing, and Chuxiong of Yunnan province, Xuchang and Luoyang of Henan province, and Qianxinan of Guizhou province (Figure 5, Table 2). Among these eight species, *Ochrobactrum anthropi* from Mangdui and *Pantoea agglomerans* from Chuxiong have been previously reported as strains capable of heavy metal detoxification [42]. *Leclercia adecarboxylata* was previously reported as a PAH-degrading strain. *Enterobacter ludwigii* from Menglijiaojidi is capable of crop-growth promotion and alkane degradation [62,63]. *Ochrobactrum lupini* from Qujing was known to degrade pesticides (chlorothalonil and cypermethrin) [50,51]. The other three species identified in the LEfSe have no previously reported functional associations, but their genera do contain species capable of degrading PAHs and heavy metals detoxication [64,65,66,67,68,69]. For instance, *Enterobacter* sp. PRd5 can reduce pyrene [64]; *Methylobacterium hispanicum* EM2 can remove Pb(II) [67]; and *Sphingomonas* sp. gy2b was reported as a PAH-degrading strain [69].

We explored the potential local factors that have potentially fostered the high relative abundance of these locally specific contaminant-degrading strains, specifically by measuring heavy metal concentrations of (Cd, Cr, Cu, Pb, and Zn) of six samples (Mangdui, Menglijiaojidi, Qujing, Chuxiong, Luoyang, and Qianxinan) as well as another 10 samples as references. Compared with other samples (including samples from the same province), Mangdui had the highest the Cd concentration, Chuxiong had the highest Cr concentration, and Luoyang had the highest Pb concentration, while Qujing had the second-highest Cu concentration of any site, and it showed significantly higher Zn concentration among the detected samples (Figure 5).

## 4. Conclusions

In this study, we systematically examined the national diversity of bacterial communities on leaves of a single crop with 16S rDNA sequencing technology, and we found a total of 1234 species (4473 OTU) from 78 samples in eight provinces of China. We determined the core bacterial community of a nation (China) and the core communities for eight Chinese provinces, which can facilitate future efforts to study crop microbiomes across geographically diverse distributions. Our findings suggest that local factors drive the microbial community composition in the local environment and emphasize that there are many potentially useful functional bacterial resources present on tobacco leaves that could be harnessed for a variety of bioprocess, agricultural, and environmental detoxification applications.

## Figures and Tables

**Figure 1 microorganisms-10-00450-f001:**
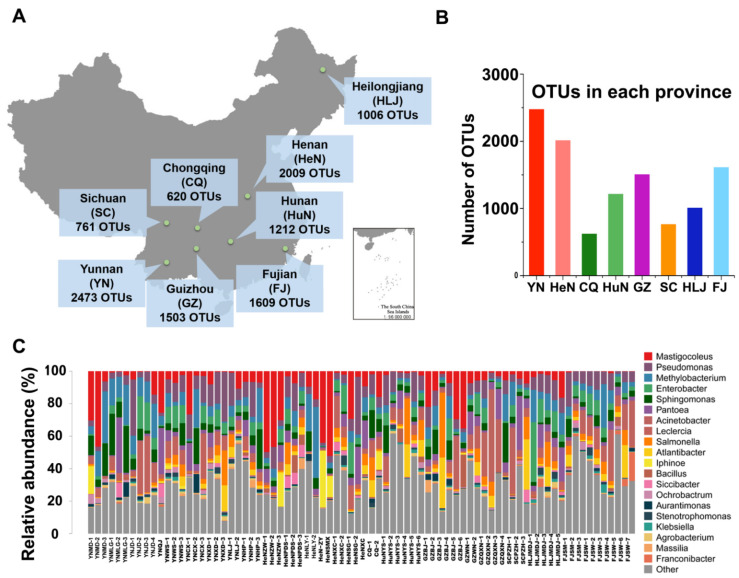
Sampling and initial phylogenic distributions. (**A**) Sampling locations of the present study. (**B**) Number of OTUs in each province. YN, Yunnan province; HeN, Henan province; CQ, Chongqing city; HuN, Hunan; GZ, Guizhou province; SC, Sichuan province; HLJ, Heilongjiang province; FJ, Fujian province. (**C**) The community compositions at the genus-level. The relative abundances of the following genera were >1%: *Mastigocoleus*, *Pseudomonas*, *Methylobacter*, *Enterobacter*, *Sphingomonas*, *Pantoea*, *Acinetobacter*, *Leclercia*, *Salmonella*, *Atlantibacter*, *Iphinoe*, *Bacillus*, and *Siccibacter*.

**Figure 2 microorganisms-10-00450-f002:**
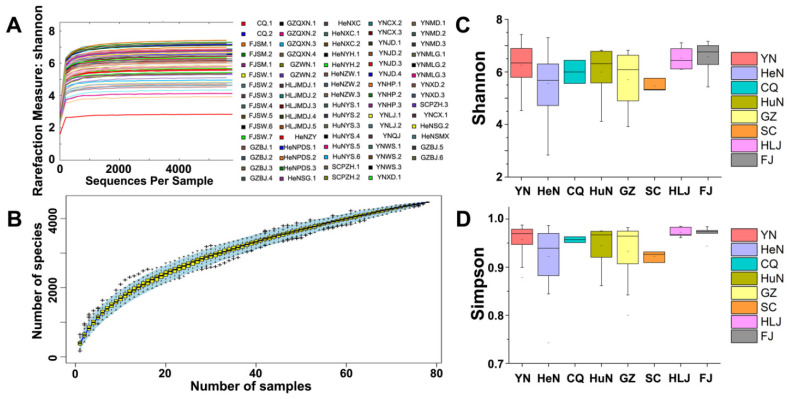
*α*-diversity analysis. (**A**) The Shannon rarefaction curves of 78 samples. The curve has reached a plate stage, indicating that the sequencing depth of the samples is suitable for further study. (**B**) Species accumulation curves of 78 samples. The curve tended to be flat, indicating that the bacterial richness of the 78 samples is sufficient for further analysis. (**C**,**D**) Box plots of the Shannon and Simpson index values of YN, HeN, CQ, HuN, GZ, SC, HLJ, and FJ province, respectively. Microbial diversity was highest in Fujian province and lowest in Henan province.

**Figure 3 microorganisms-10-00450-f003:**
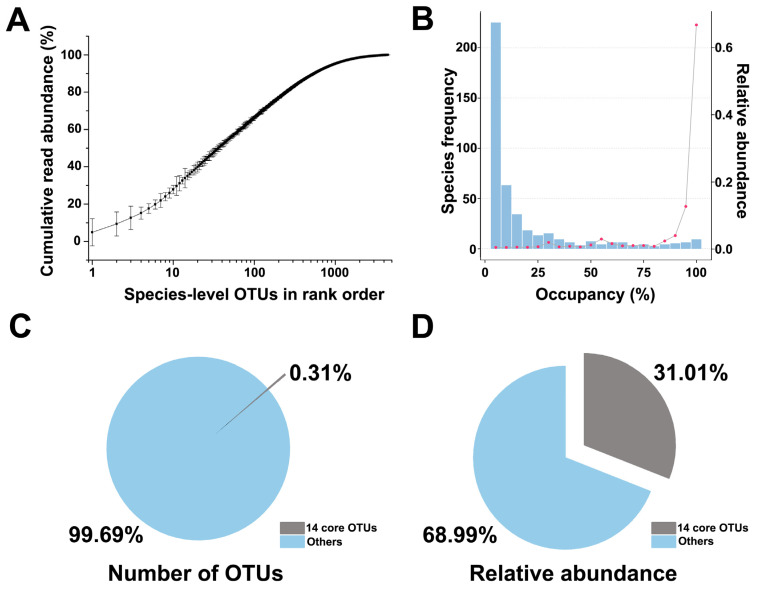
Core bacterial community. (**A**) Cumulative read abundance across the 78 samples. Cumulative read abundance (mean ± SD) of species-level OTUs plotted in rank order for the 78 samples. In each sample, the 10 most abundant OTUs made up 27.78% (±2.26%) of the total reads on average, and the 100 most abundant OTUs made up 66.32% (±0.34%). OTUs were considered “abundant” in a sample when they were among the OTUs comprising the top 80% of reads. (**B**) Frequency and abundance distributions of all OTUs. OTUs were classified into bins according to their occupancy of samples. For example, the first bin contains OTUs detected in fewer than 5% of the samples, while the last bin contains OTUs detected in 95–100% of the samples. Species frequency (the number of OTUs) is indicated by the blue bars; the total abundance of OTUs in each bin is indicated by the red points. (**C**,**D**) Percentage and relative abundance of the core OTUs versus the remaining microbial OTUs. In total, 0.31% (14 of 4473 OTUs) OTUs were identified as abundant and frequent at the countrywide scale; these OTUs accounted for (on average) 31.01% of the 16S rRNA gene sequences of all samples.

**Figure 4 microorganisms-10-00450-f004:**
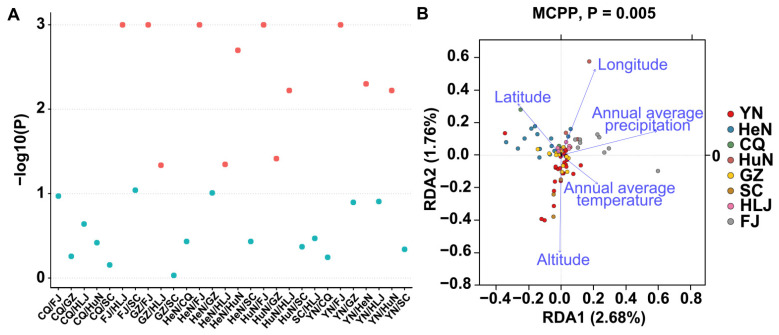
*β*-diversity and redundancy analysis. (**A**) Multi Response Permutation Procedure (MRPP). The composition difference between each province was valued by −log10 (*p*). The red dots are the pairs that differ significantly, while the blue dots are those that do not differ. More than half of the pairs showed significant differences, indicating that the microbial community composition varies greatly from province to province. (**B**) Redundancy analysis. The relationship between the microbial diversity and temperature, precipitation, altitude, longitude, and latitude of each sample was analyzed. Each arrow represents an environmental factor, with each dot representing one sample. An acute angle between arrows indicates a positive correlation, an obtuse angle indicates a negative correlation, and a right angle indicates no correlation.

**Figure 5 microorganisms-10-00450-f005:**
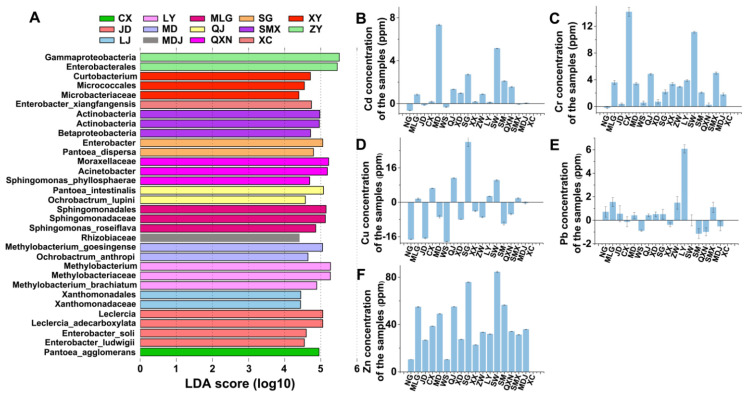
LEfSe analysis and heavy metal concentration determination. (**A**) LEfSe (linear discriminant analysis effect size). The ordinate is the taxon that differs significantly between groups; the abscissa visually shows the LDA difference analysis logarithm score (LDA > 2, *p* < 0.05) for the corresponding taxon (as a bar graph). (**B**) The concentration of Cd. (**C**) The concentration of Cr. (**D**) The concentration of Cu. (**E**) The concentration of Pb. (**F**) The concentration of Zn. Negative values refer to no ions present. NG, negative group; CX, Chuxiong; LY, Luoyang; MLG, Mengligong; SG, Shigang; XY, Xiangyang; JD, Menglijiaojidi; MD, Mangdui; QJ, Qujing; SMX, Sanmenxia; ZY, Zhuyang; LJ, Lijiang; MDJ, Mudanjiang; QXN, Qianxinan; XC, Xuchang. WS, Wenshan; XD, Xindian; ZW, Zhangwu; SW, Shaowu; SM, Sanming.

**Table 2 microorganisms-10-00450-t002:** The significantly different species among the samples.

Sampling Location	Species	Function
Yunnan-Mangdui	*Ochrobactrum anthropi*	Adsorption of heavy metal ions [49]
Yunnan-Menglijiaojidi	*Leclercia adecarboxylata*	Degradation of PAHs [45]
Yunnan-Menglijiaojidi	*Enterobacter ludwigii*	Promotion of plant growth, alkane degradation [62,63]
Yunnan-Qujing	*Ochrobactrum lupini*	Degradation of pesticides (chlorothalonil and cypermethrin) [50,51]
Yunnan-Chuxiong	*Pantoea agglomerans*	Heavy metals detoxification [61]
Henan-Xuchang	*Enterobacter xiangfangensis*	Function unknown, but the corresponding genus is able to degrade PAHs and heavy metals [64,65]
Henan-Luoyang	*Methylobacterium brachiatum*	Function unknown, but the corresponding genus is able to degrade PAHs and heavy metals [66,67]
Guizhou-Qianxinan	*Sphingomonas phyllosphaerae*	Function unknown, but the corresponding genus is able to degrade PAHs and heavy metals [68,69]

## Data Availability

All raw data generated in this study are included in the Appendix A. We also added the original sequencing data to NCBI’s database (Submission ID: SUB11032938, BioProject ID: PRJNA803016).

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
