# Peer review of "The Leaf Microbiome of Tobacco Plants across Eight Chinese Provinces"

_microorganisms, 2022, doi:10.3390/microorganisms10020450_

Round 1

Reviewer 1 Report

Major comments:

The paper is poorly structured - the methodology lacks information about all methods of statistical analysis and data visualization; they are mentioned only in the results. The methods used are not supported by any references. In the results the authors often write that they have done some analyses etc., but they do not mention their own results, they only refer to tables, charts. Without discussing the results. In the meantime, in subsection 3.5, elements of discussion appear. In the actual discussion, however, there is no such discussion...

Other comments:

L20: explain LEfSe

L63: explain OTUs

L64-69: the summary should not be in the introduction; this is the place for the aim and objectives of the study

L72: give these locations, GPS, soil types, types of crops from which plants were taken etc. can be in a supplement, but this is important info. Also - how were the plants selected, when were the samples taken, how were the leaves taken, what leaves? There is no info about climate, rainfall, soil type....

L77-82: references to methodology; i.e. microorganisms were analysed from the whole leaf from the outside? Not from the inside?

L85-86: references to primers

L87-93: references to methodology

L95-99: references to methodology

L95: UCLUST - from the information on this algorithm/program it appears that it should not be used for designed for OTU clustering. Why did the authors use this method? Any literature? https://drive5.com/usearch/manual/uclust_algo.html

L99: sequences have been added to the database? if so please provide a draft, if not please do so

L101-104: references to methodology

L115: references to R

L122-126: references

L128-161: references to methodology!

L170: information on methods and tools used for visualisation should be included in the methodology

L178-184: this should be in the methodology, not in the results

L189: This should be in the methodology, not in the results

L211-213: this should be in the methodology, not in the results

L240-260: this looks like a discussion and we are in the results section

L263: PcoA is not in the methodology

Figure 2A - illegible legend

L322: at what taxonomic level is richness and diversity labelled?

Figure 5B-F: X-axes illegible

Table 1, 2: on what basis were the functions of the individual bacteria determined (third column)

Discussion: this is not a discussion! We have only 1 reference here!

Reviewer 2 Report

The authors have carried out an exhaustive study on the leaf microbiome of tobacco plants located in sites with different environmental conditions. In that sense, the work is interesting and novel and, in general terms, it is well written. However, some modifications could be included that, in my opinion, could improve it.

First, the title does not correctly reflect the content of the paper. There is no mention of the plant species used or the area of the plant studied, which gives a rather more global impression of the study. In this sense, I think it would be more appropriate to specify that the analysis focuses on the leaf microbiome of tobacco plants.

Secondly, nowhere in the article is the study conducted in relation to heavy metals substantiated. The data are simply presented without any justification for their presence.

The discussion is excessively brief, while certain paragraphs located in the results section would correspond more to the discussion. The results should be descriptive, without entering into evaluations or comparisons with those obtained in other studies.

Regarding the number of samples, the authors mention a total of 78, corresponding to 3 replicates from 26 locations. The minimum number of samples per location is said to be 3, being higher in 16 locations. If the latter were true, the number of samples would be higher than 78.

As described by the authors, the most abundant sequences in each OTU were chosen as representative. Why was a consensus sequence not used?

In section 2.13., the units must be separated from the parameter value. Additionally, "diluted" should be changed to "dilute".

Reviewer 3 Report

A report for: microorganisms-1566827 A Reference Map of Microbiome Composition in Crop Produc- 2 tion across 8 Chinese Provinces  

It is an interesting paper of high importance in which authors profiled the microbiomes of 78 primary dried tobacco leaf samples from plants grown at 26 locations in 8 Chinese provinces. The result and discussion was in-depth. Statistical tools were used, and comparison with previous work was descriptive and qualitative. The authors stated there are significant differences in microbial community structures among provinces and clear clustering by región.  In my opinion, the manuscript needs some work to become suitable for publication.

-Please, the objectives must be explicitly clear.

-I think that the abstract is a very important part of the manuscript, after reading the manuscript I suggest to rewrite, at less some sentences.

-The materials and methods are exposed in a cumbersome way, please improve the content. In Line 72. Crop leaves were collected from 26 locations in China. A total of 78 samples were collected from 26 sampling locations in China. Please rewrite and give more details.

-Please improve the figure 1, 2, 4 and 5.

-I susggest to include some photos of the sampling and the areas

-Conclusions are necessary and essential.

I wish those changes will contribute to improve your paper.

Round 2

Reviewer 1 Report

Thank you for your corrections and replies to comments. I still have comments.

L99/100: comments „Yes, we have already added the original data to NCBI’s database. We get the Submission ID: SUB11032938, and BioProject ID: PRJNA803016.” this information must be included in the manuscript.

L117: the version of R used must be indicated.

L125: reference to UPGMA

L184-210; 253-299: no discussion

Figure 1B - and the figure is OUTs instead of OTUs

Figure 1C - not very readable, why isn't it just a separate figure?

Figure2C and D - maybe it is worth changing the unit on Y axis so that the axis starts from e.g. 2.0 in fig. C, and 0.7 in fig. D. Graphs will be clearer.

Table 1 and 2: second column "Specy"? Should be "Species"

Table 2: “Function unknown, but the corresponding genus is able to degrade PAHs and heavy metals” please provide reference, give example.

Reviewer 3 Report

All the comments and suggestions that I mentioned have been responded and revised; then the paper should be accepted for publication now.

Author Response

Thanks again for handling our manuscript, and helpful comments on our manuscript.